# Influence of the Chitosan and Rosemary Extract on Fungal Biodegradation of Some Plasticized PLA-Based Materials

**DOI:** 10.3390/polym12020469

**Published:** 2020-02-18

**Authors:** Elena Stoleru, Cornelia Vasile, Lăcramioara Oprică, Onur Yilmaz

**Affiliations:** 1Physical Chemistry of Polymers Department, “Petru Poni” Institute of Macromolecular Chemistry, 41A, Grigore Ghica Voda Alley, 700487 Iasi, Romania; elena.paslaru@icmpp.ro; 2Faculty of Chemistry, “Alexandru Ioan Cuza” University of Iasi, 11 Carol I Blvd, 700506 Iasi, Romania; 3Biology Faculty, “Alexandru Ioan Cuza” University of Iasi, 11 Carol I Blvd, 700506 Iasi, Romania; 4Leather Engineering Department, Faculty of Engineering, Ege University, 35100 Bornova/Izmir, Turkey; onuryilma@gmail.com

**Keywords:** PLA, biodegradation, biochemical parameters, thermal behavior, bioactive compounds

## Abstract

The fungal degradation of the complex polymeric systems based on poly(lactic acid) (PLA) and natural bioactive compounds (chitosan and powdered rosemary alcoholic extract) was studied. Two fungal strains, *Chaetomium globosum* and *Phanerochaete chrysosporium* were tested. Both fungi characteristics and changes in morphology, structure and thermal properties were monitored. Biochemical parameters as superoxide dismutase, catalase, soluble protein and malondialdehyde have been determined at different time periods of fungal degradation. The fungi extracellular enzyme activities are slightly decreased in the case of composites containing bioactive compounds. The presence of natural compounds in the PLA-based polymeric system determines an acceleration of fungal degradation and probably the chemical hydrolysis, which further helps the attachment of fungi on the surface of polymeric samples. Significant decreases in average molecular mass of the polymeric samples were observed by fungi action; accompanied by structural changes, increase in crystallinity and decrease of thermal properties and the loss of the physical integrity and finally to degradation and integration of fungal degradation products into environmental medium. It was found that both fungi tested are efficient for PLA-based materials degradation, the most active from them being *Chaetomium globosum* fungus.

## 1. Introduction

The lack of degradability and xenobiotic nature of the synthetic polymeric materials led to high levels of environmental pollution and health hazards. Recalcitrant nature of plastic is a matter of huge environmental concern, and hence new challenges are related to plastic degradation. With the increasing demand for plastics and rising pressure for their safe disposal, biodegradable plastics and plastic biodegradation gained a lot of attention in the last years. Plastics biodegradation is one of the best possible ways to treat these recalcitrant materials in an environment-friendly manner. The best possible routes of biodegradation and their associated social and environmental impacts must be selected [1]. The application of biopolymers is limited to short shelf-life products, but they show a huge potential to govern the packaging sector in the future [2].

The environmental degradation process and its duration and mechanism strongly depend on the environmental conditions including heat, humidity, pH, oxygen, microbes and so on. High-temperature environments with a rich microorganisms presence facilitate degradation. Poly(lactic acid) (PLA) biodegradation is a very controversial problem because it requires temperature around 60 °C (PLA glass transition temperature) which is not reached in normal soil conditions. In a regular room environment, PLA will resist for many years. However, PLA biodegrades because its origins are natural, is a bio-based polymer, the monomers being derived from renewable resources [3] and microorganisms can feed on it to turn it into compost [4,5]. Humidity, 60 °C temperature and microorganisms are required for PLA biodegradation. Occasionally such conditions are found in garden soil. In good conditions, PLA will show signs of biodegradation in 6 months [6]. Enzymes which hydrolyze PLA are not available in the environment except on very rare occasions. Proteinase K catalyzes the hydrolytic degradation of PLA but it is not commonly present in natural environments [7,8].

The knowledge of the biodegradation behavior of polymers is crucial for treating plastic wastes and the serious energy crisis. The analysis of biochemical processes involved in PLA biodegradation is a key factor for exploring the high efficient methods of PLA biodegradation. PLA degradation occurs mainly through scission of ester bonds into carboxylic acid and alcohol. In addition, polymer degradation is induced by a range of factors in nature, such as oxidation, photodegradation, thermolysis, hydrolysis, biodegradation or enzymolysis [9,10]. During microbial degradation, PLA-degrading microorganisms first excrete extracellular depolymerase of PLA simulated by peptides and proteins [11]. The depolymerase attack intramolecular ester links of PLA, which result in production of oligomers, dimers and monomers. Afterwards the low molecular weight compounds enter in microbial membranes and decompose into carbon dioxide, water or methane by intercellular enzymes [12,13].

Therefore, the researches on pure isolation of PLA-degrading microorganisms have been raised in recent years. Currently, multiple types of microbes that are able to degrade PLA have been isolated from soil or water. They mostly are actinomycetes (*Amycolatopsis, Saccharothrix, Lentzea, Kibdelosporangium*, *Streptoalloteichus*, *Pseudonocardiaceae)* which accelerate PLA biodegradation in the natural soil microcosms [14,15], bacteria (*Bacillus, Pseudomonas, Stenotrophomonas*, etc.) and various fungi [10]. Several efficient methods were developed to accelerate degradation of commercial polylactic acid as: UV-C treatment of PLA beverage cup which reduced the molecular weight of PLA and also of the polyethylene dumped in the environment because of the presence of the *Bacillus subtilis* [16] which produces surface active compounds (e.g., Biosurfactants). Moreover, the burial in soil amended with various sources of microbial consortium, including cow-manure compost, green yard-waste compost, and wastewater sludges from dairy, rice vermicelli and coconut milk factories, dairy wastewater sludge accelerated complete PLA biodegradation after 15 days. A combined use of dairy sludge and *Pseudomonas geniculata*—WS3 is recommended for accelerating PLA degradation [17].

A few studies reported in the scientific literature deal with the fungal degradation of PLA. There are 14 filamentous fungal strains (*Fusarium moniliforme* and *Penicillium roqueforti*) able to assimilate d,l-lactic acid in liquid cultures. Moreover, fungal degradation of PLA had also been studied by using *Tritirachium album* (ATCC 22563) in a liquid culture [18]. It shows that most of PLA film is degraded after 14 days of cultivation by the addition of gelatin. Many studies mainly focused on the degradation of PLA by the pure cultures of fungi. However, some researchers have explored the fungal degradation of PLA in soil and compost [19,20]. 

Recent scientific studies on the microbial degradation of natural and synthetic molecules show the potential of fungal application on cleaning technologies. It was showed by Vivi et al. [21] that from all utilized fungi, *Chaetomium globosum* (ATCC 16021) was a pioneer in the colonization and attack of polycaprolactone (PCL) resulting in significant mass losses. *C. globosum* is a saprophytic basidiomycete fungus with high capability on degrading plant materials having cellulolytic enzyme systems [22]. *Chaetomium globosum* is a well-known mesophilic member of the mold family Chaetomiaceae. This saprophytic fungus primarily resides on plants, soil, straw and dung [23]. The basidiomycete *Phanerochaete chrysosporium* is a model of ligninolytic fungus which has been studied for a long time [24]. The production of reactive oxygen species (ROS) by this fungus occurs in physiological conditions, as during the wood degradation. In our previous paper, *Phanerochaete chrysosporium* white rot fungus was tested for its ability to biodegrade poly(lactic acid)-based materials gamma irradiated or nitrogen plasma activated and chitosan-surface grafted [25]. The biodegradation of the PLA-based systems by *Trichoderma viride* fungus, in liquid medium and controlled laboratory conditions was also studied [26]. In both cases the accelerated fungal degradation of PLA based materials was found accompanied by average molecular weight decrease, structural and morphological changes and deterioration of the properties. 

In order to gain valuable insights into the biodegradation of complex polymeric systems, the present paper aimed to develop a comparative study of the action of *C. globosum* and *Phanerochaete chrysosporium* fungi on the degradation of PLA-based complex materials containing 20 wt% polyethylene glycol (PEG) as plasticizer, 3–6 wt% chitosan (CS) and 0.5 weight percent (wt%) rosemary extract (R). It is expected that the bioactive compounds CS and R to influence the fungal degradation, as they exhibit antimicrobial, antioxidant activities and show a high potential application for food preservation and for therapeutic beneficial use, because they “contain active ingredients parts of plants, or other plant materials, or combinations” [27,28].

The biodegradation was evaluated from two points of view: following the influence of the polymeric materials on biochemical enzymes specific for fungal metabolism and also by changes in physical-chemical properties of the bio-based polymeric materials as changes in average molecular weight, morphology, structure and thermal behavior.

## 2. Materials and Methods

### 2.1. Materials

PLA-based materials subjected to biodegradation under the action of *Chaetomium globosum* (CG) and *Phanerochaete chrysosporium* (PC) fungi were PLA and polyethylene glycol (PEG)-plasticized PLA as reference samples, and biocomposites containing 0.5 wt% powdered rosemary ethanolic extract (R), 3 wt% and 6 wt% chitosan (CS) and both additives—Table 1. The natural bioactives as chitosan and rosemary ethanolic extract were selected to offer multifunctionality, as antimicrobial, antioxidant and biocompatibility to PLA materials, with potential applications both in food packaging and biomaterials, as it was demonstrated in our previous paper [29]. Their role was described in our previous papers [26,30].

### 2.2. Fungal Material

Detailed experimental conditions have been described in a previous paper [25]. *Chaetomium globosum* and *Phanerochaete chrysosporium* fungi purchased from the Institut Scientifique de Sante Publique, Belgium, preserved in the laboratory conditions have been used in this study. Peptone, yeast extract, malt extract, agar-agar and Coomassie Brilliant Blue G250 (Fluka Switzerland), nitro blue tetrazolium, riboflavin, o-dianisidine and hydrogen peroxide (Sigma-Aldrich, Germany), sulfuric acid, perchloric acid, 2-thiobarbituric acid, bovine serum albumin and other chemicals (Merck, Germany) were of analytical grade ones. High purity double distilled water was employed for preparation of solutions.

### 2.3. Fungal Strains and Culture Conditions 

*C. globosum* was grown on agarized Haynes medium (yeast extract 4 g/L, malt extract 10 g/L, glucose 4 g/L, agar 15 g/L in distilled water) [31], while *P*. *chrysosporium* fungus was cultivated with solid Sabouraud agar growth medium (peptone 10 g/L, glucose 35 g/L, agar 20 g/L in distilled water), keeping them for 7 days at 28 ± 0.1 °C and stored at 4 °C. The discs of 0.8 cm in diameter from both fungi cultures were inoculated into 100 ml Haynes liquid or Sabouraud liquid media with 7 days old cultures of fungi, respectively using Erlenmeyer flasks. The biodegradation test was performed in controlled laboratory conditions for 14 days using the polymeric samples cut into 20 × 20 mm specimens, thickness 3 ± 0.1 mm. Samples were incubated at 28 ± 0.1 °C in the Incucell room and grown to stationary phase. The both controls were represented by the mycelium grown in the absence of polymeric samples. 

### 2.4. Investigation Methods

#### 2.4.1. Scanning Electron Microscopy (SEM)

The surface morphology of the films before and after fungal biodegradation was examined by scanning electron microscopy (SEM), using a QUANTA 200 scanning electronic microscope (FEI Company, Hillsboro, OR, USA), at an accelerating voltage of 10 and 20 kV. The surfaces of PLA-based samples were examined as such (without metal coating) and SEM micrographs being recorded at different magnifications (given on the images). This technique allows to directly performing morphological and surface characterization as well as their three-dimensional shape of morphological features, the presence of aggregates, voids, etc. SEM is a useful tool for evaluating the changes that occur in the microstructure of polymers during degradation/ageing.

#### 2.4.2. Biochemical Assays

To *C. globosum* or *P. chrysosporium* mycelium (0.5 g) were added 2 mL of 0.1 M phosphate buffer solution (pH 7.5) and were subsequently centrifuged at 11,300 rpm, at 4 °C. The superoxide dismutase, catalase, soluble protein and malondialdehyde, as biochemical parameters were determined on fungus mycelium after different inoculation period as 7 and 14 days. Superoxide dismutase (SOD) activity was obtained from light absorbance at 560 nm wavelength by adapted colorimetric Winterbourn method, while the catalase (CAT) activity through the method described by Sinha (1972) [32]. Lipid peroxidation was determined as malondialdehyde (MDA) content using thiobarbituric acid (TBA) according to the procedures described by Hodges et al. [33]. All results were expressed relatively to protein content (according to Bradford, 1976 assay) [34] and graphically drawn as average values and standard deviations resulted from five repeated measurements.

#### 2.4.3. ATR-FTIR Spectroscopy

To determine the structural changes occurred during fungal degradation of the PLA-based biocomposites, the attenuated total reflection-Fourier transform infrared spectroscopy (ATR-FTIR) technique was used. The background and samples spectra were recorded with a Bruker VERTEX 70 spectrometer (Bruker, USA) in the 600 to 4000 cm^−1^ wavenumber range, at a 4 cm^−1^ resolution by performing 136 scans. The ATR module is equipped with a 45° ZnSe crystal (penetration thickness is around 100 µm). For each sample, the evaluations were made on the average spectrum obtained from three recordings. Spectrum processing was performed using OPUS and ORIGIN programs. 

#### 2.4.4. Gel Permeation Chromatography (GPC)

Average molecular weights of the samples before and after incubation with fungi were determined by Gel Permeation Chromatography instrument WGE SEC-3010 multi-detection system (Brookhaven, GA, USA), consisting of a pump, two PLgel columns (PLgel is a highly cross linked, porous polystyrene/divinylbenzene matrix), and dual detector RI/VI (Refractometer/Viscometer) WGE SEC-3010. The working solutions were prepared in chloroform (CHCl_3_) and the experiments were achieved using a flow rate of 1.0 mL/min at 30 °C temperature. The RI/VI detector was calibrated with PS standards (580-1350000 DA) having narrow molecular weight distribution. Fragments of the samples were dissolved in HPLC grade chloroform (CHCl_3_) (0.02 g/mL CHCl_3_) and stirred at 25 °C, for 1 h, using a heater-stirrer. Diluted solutions were filtered using a 0.2 Teflon filter to extract insoluble fraction, before injecting the solute through the PLgel columns. Weight average molecular mass (Mw), number average molecular weight (Mn), as well as the polydispersity index (Mw/Mn) and viscosity were determined, using the PARSEC Chromatography software. The mean of at least two injections are given.

#### 2.4.5. Differential Scanning Calorimetry (DSC)

For Differential Scanning Calorimetry a Shimadzu 60 Plus instrument (Kyoto, Japan) was used to examine the thermal behavior of PLA-based samples. The samples were sealed in aluminum pans. Differential scanning calorimetry (DSC) analysis was conducted under nitrogen flow (20 mL/min) within a temperature range of 20 and 250 °C with a heating rate of 10 °C/min. Calibration was performed using an indium standard (*T*_m_ = 156.6 °C; Δ*H*_m_ = 28.45 J/g). The degree of crystallinity (*X*_c_) of the PLA and its composites was calculated by dividing the melting enthalpy of the sample by Δ*H*_m_ = 93.7 J/g, which is equilibrium enthalpy of a PLA sample with 100% crystallinity (PLA 100%). An overall accuracy of ±0.5 °C in temperature and ±1% in enthalpy was estimated [35].

#### 2.4.6. Thermogravimetry (TG)

Thermogravimetry was performed by using a Perkin Elmer TGA 8000 (Shelton, CT, USA) device to evaluate the thermal degradation process of the samples. The temperature program was set from 30 to 700 °C with a heating rate of 10 °C/min under N_2_ flow (20 mL/min). 

## 3. Results and Discussion

### 3.1. Visual Inspection and Scanning Electron Microscopy (SEM)

The PLA-composite samples were characterized before and after each exposure period (7 and 14 days) in fungal media.

The presence of biodegradable polysaccharide, chitosan, into PLA-based composites hastened the disintegration rate; the composite samples exposed to liquid incubation medium were completely disintegrated after 7 days. Is noticed that the presence of PEG plasticizer in PLA matrix leads to an accelerated degradation of the polyester, the visual inspection highlighting the changing color from pale yellow (initial PLA/PEG) to white and a marked increase of brittleness (after 14 days the samples become difficult to handle)—Figure 1. It is also possible that PEG is lost from samples during fungal test.

The occurred changes in the optical properties of the samples may be due to the modification of their crystallinity and/or water absorption and to biodegradation process. Based on visual examination it can be noted that the samples exposed to *Chaetomium globosum* fungus undergoes a more pronounced change of the initial features.

The SEM photomicrographs (Figure 2) show that the surface of non-degraded PLA material is smooth, without cracks and holes, while after fungal exposure the surfaces of PLA and PLA/R become rougher, being highlighted surface defects. Is noticed also that the samples were covered with hyphae and characteristic spores, meaning that the used fungi expanded their colonies and cover the entire samples surface, this aspect being more pronounced in the case of exposure to *Chaetomium globosum* fungus. The presence of the rosemary powder into PLA matrix composition, besides the adhesion of the fungi to the surface, causes cracks into the sample.

Dense network of fractures was particularly visible for the samples containing PEG; this may be due to the hydrophilic nature of the plasticizer, which makes the samples more accessible for fungal attack. PLA/PEG and PLA/PEG/R samples after 14 days incubation with *Chaetomium globosum* and *Phanerochaete chrysosporium* fungi were covered by cracks of varying diameters which eventually created holes. Larger cracks are observed when the samples were incubated with *Chaetomium globosum*, resulting in loss of integrity of the sample. These findings may suggest that PLA-based samples served as the source of carbon and energy for the studied fungi. As revealed by SEM images in Figure 3, peeling and exfoliation appeared in sample containing PEG and rosemary powder, the destructive process of biodegradation being most prominent in this case. The SEM micrographs highlights that the fungal degradation clearly occurs from the surface.

### 3.2. Weight Loss Measurements

Weight losses are widely applied in the biodegradation tests, although no precise proof of evaluation can be obtained [36]. In the present study, the weight loss was determined after 7 and 14 days of fungus incubation, by carefully removing the sample from the medium and washing it with distilled water. The samples were subsequently dried at 30 °C, until constant weight was obtained. The weight losses (Equation (1)) were calculated as difference between the mass of the sample after biodegradation (*m_d_*) and before (*m*_0_) and were expressed as percentages.
(1)$$Weight\;loss\;\left(\%\right)=\frac{m_{0}-m_{d}}{m_{0}}\times 100$$


Measurements of mass loss can be problematic due to moisture absorption or difficult recovery of disintegrated material. In Table 2 is shown the percentage weight loss of the samples degraded by *Chaetomium globosum* and *Phanerochaete chrysosporium*, the values represent an average of three experiments. 

It is known that native PLA has a slow rate of disintegration and the microorganisms that degrade PLA are rarely found in nature [37]. Among the tested samples, the native PLA showed the lowest degradation rate, only 22% weight loss was found after 14 days of exposure to *Chaetomium globosum*. The PLA-based composites samples exhibited relatively high rates of weight loss, 43% within 14 days. In the presence of the rosemary extract alone, the weight loss is smaller mainly after 7 days fungal degradation because of its high content in phenolic compounds with antioxidant activity [38]. The PEG plasticized PLA-based composites and containing CS and R were the most rapidly degraded samples, which exhibited a 100% weight loss (more exactly the recovery of disintegrated material was not possible or they are totally integrated in fungal degradation media). The sterile controls (polyethylene films) showed no weight loss in the tested period. The addition of hydrophilic plasticizer (PEG) caused an increase in weight loss of 43–57%. The rate of weight loss differed between the studied fungi cultures, the samples inoculated with *Chaetomium globosum* shows average percentage of weight loss higher than the ones exposed to *Phanerochaete chrysosporium*. Similar results were reported also by Geweely and Ouf [39] testing these two fungi on starch based polymers. 

### 3.3. Biochemical Results

Both the characteristics of the fungi and their activities on polymer degradation depend on enzymes action, therefore, biochemical studies are necessary to elucidate the fungal bioactivity which gives information about their role in polymer biodegradation. Cells continuously produce free radicals and reactive oxygen species (ROS) as part of their metabolic processes, but in case of their overproduction the oxidative stress occurs. Current knowledge has shown that ROS play a key role as an agent in normal cell signal transduction and cell cycling [40].

Hancock et al. (2001) [41] demonstrated in their work that ROS have a role in cell signaling, including apoptosis; gene expression; and the activation of cell signaling cascades. As a result of this, several barricade mechanisms have evolved to meet this need and provide a balance between production and removal of ROS. Cells have categories of fortifying mechanisms to ameliorate the harmful effects of ROS. The ROS defense mechanism consists of the antioxidant machinery which helps to mitigate the above-mentioned oxidative stress-induced damages. The damaging effects of ROS are ameliorated by different antioxidative defense systems and the antioxidant machinery has two arms with the enzymatic components and non-enzymatic antioxidants [42]. Antioxidants are compounds that inhibit oxidation process. By their radical scavenging activity, they assure the protection of the materials such as biomolecules [43,44].

To evaluate the ROS-scavenging action, in the present study, the activities of two key antioxidant enzymes, superoxide dismutase and catalase, were monitored as being of importance in oxidative stress defense in *P. chrysosporium* and *C. globosum* fungi (Figure 4 and Figure 5a,b). The enzymes work together to detoxify the ROS generated by biotic and abiotic stress [45].

Another parameter of oxidative damage evidenced in the mycelium of both fungi was measured as malondialdehyde content (MDA) (Figure 6a,b).

*Effect of the PLA-based samples on superoxide dismutase activity*. Superoxide dismutase (SOD) (EC.1.15.1.1) is considered as a key enzyme in cells, a first line of defense against oxidative damage induced by various environmental factors. The SOD catalyzes the dismutation of two molecules of superoxide anion ($O_{2}^{-}$) to hydrogen peroxide (H_2_O_2_) and molecular oxygen (O_2_), consequently rendering the potentially harmful superoxide anion less hazardous one [46]. The polymeric samples introduced in culture media of *Phanerochaete chrysosporium* determine at 7 days after incubation a diminution of SOD activity (Figure 4a). The sample with PLA/PEG/6CS/R caused a decrease of SOD activity with 50% reported to control. In *Chaetomium globosum* mycelium, SOD activity recorded a heterogeneous response; the sample with PLA/PEG/6CS/R determined the lowest enzyme activity (24%) while PLA sample the higher activity (7%) compared with control. At another studied time interval, 14 days after inoculation, the trend of SOD activity was the same, for both species of fungi but on a different scale values (Figure 4b). Thus, in case of *P. chrysosporium* the profile of SOD activity was similar with that recorded at 7 days after incubation with polymer samples; in addition, the sample with PLA/PEG/6CS/R was found to majorly contribute to the decrease of enzyme (50%) comparatively with control. On the other hand, in *C. globosum* mycelium, SOD activity was very different, PLA/PEG/R sample determined the maximum activity (stimulation rate 20%) while PLA/PEG/3CS samples evidenced a lowest activity 15% compared with the control.

*Effect of the PLA-based samples on catalase activity.* Catalase (CAT) (EC 1.11.1.6) is a common antioxidant enzyme present almost in all living tissues that utilize oxygen which catalyzes the degradation or reduction of hydrogen peroxide (H_2_O_2_) to water and molecular oxygen, consequently completing the detoxification process imitated by SOD [47]. Among all antioxidative enzymes, CAT has one of the highest turnover rates: one molecule of CAT can convert around 6 million H_2_O_2_ molecules to H_2_O and O_2_ per minute, and stress conditions reduce the rate of protein turnover [48].

The data of Figure 5a indicated that at 7 days after inoculations, the majority of polymeric samples (excepting PLA/PEG) determined a decrease of CAT activity; in *P. chrysosporium* mycelium a more intense decrease (41%) was recorded at PLA/PEG/6CS/R sample. The CAT activity was diminished by introducing the polymeric samples in culture media of *C. globosum* and the lowest value being registered at PLA/PEG/3CS/R sample. In the 14 day after inoculation (Figure 5b) of *P. chrysosporium* media, all used polymers caused a decrease of CAT activity, the lowest activity being at PLA/PEG/6CS. In *C. globosum* mycelium, the CAT activity was lower than control. However, only polymers PLA/PEG/6CS and PLA samples had CAT values comparable to those of the control.

An interesting finding is the fact that after 7 days of incubation with fungi, the presence of PEG plasticizer into PLA matrix determines an increase of enzymes activity, while after 14 days this feature is no longer preserved. Evaluating the extracellular enzyme activity in mycelium for both studied fungal strains, one can conclude that the presence of chitosan and rosemary powdered extract into PLA-based samples led to the consumption of the enzymes produced by the two fungi in biodegradation process of the PLA-based complex polymeric samples. The action of the *C. globosum* seems to be higher than that of *P. chrysosporium.* This is not really surprising taking into account the known property of chitosan to inhibit other enzymes activities [49]. These findings suggest that biodegradation of the complex PLA polymeric samples is based on a fungal enzymatic action but also other chemical processes may play a significant role. 

*Effect of the PLA-based samples on malondialdehyde concentration.* Malondialdehyde (MDA) is one of the most known secondary products of lipid peroxidation, and it can be used as one of the most frequent biomarker of cell membrane injury [50].

The ROS-induced peroxidation of the lipid membranes is a reflection of stress-induced damage at the cellular level [51].

After 7 days, all polymer samples in contact with *P. chysosporium* media displayed a diminution of MDA content, compared to the control—Figure 6a. Thus, the low MDA content provides evidence of lower mycelium membrane damage. In contrast, the situation however changed in the case of values obtained after 14 days of experiment (Figure 6b) when all values of this biochemical parameter are much lower than those recorded after 7 day fungal degradation period up to three times smaller (about 10 nmol/mg protein in comparison with 35 nmol/mg protein) with the highest values for degradation under *P. chysosporium fungus.* Thereby, as light enhanced level of lipid peroxidation, as indicated by the MDA content, was observed after 14 days in *P. chysosporium* mycelium in response to presence of the polymeric samples. Thus, it would indicate a decreased oxidative stress as effect of polymeric samples. However, only PLA/PEG/3CS and PLA/PEG/6CS has determined the least amount of MDA measured, with 10% and 8%, respectively (Figure 6).

The polymeric samples in contact 7 days with *C. globosum* suggest MDA values very close to the control. However, the higher increase (11%) was observed in the case of PLA/PEG/6CS/R sample. Regarding the content of MDA measured after 14 days of inoculation (Figure 6b) the results showed an increase of this marker of lipids peroxidation in the mycelium of *C. globosum,* varying between 14% and 77% at PLA/PEG and PLA/PEG/3CS/R, respectively. Moreover, the polymer samples also determined at *C. globosum* mycelium an increase in MDA content, indicating the presence of oxidative stress, similar to *P. chysosporium*.

Based on the MDA level found for both fungi it can be specified that *C. globosum* fungus is more active against oxidative stresses induced by the biodegradation of complex PLA-based samples.

### 3.4. GPC Results—Average Molecular Weight Determination

Biodegradation efficiency achieved by the microorganisms is directly related to the key properties such as molecular weight and crystallinity of the polymers, which also changed during degradation, being important proofs for degradation progress. Enzymes engaged in polymer degradation initially are outside the cell and are referred to as exo-enzymes having a wide reactivity ranging from oxidative to hydrolytic functionality. The exo-enzymes generally degrade complex polymer structure to smaller, simple units that can take in the microbial cell to complete the process of degradation. Even, the major commercial polymers such as polyolefins, poly(vinyl chloride, poly(ethyleneterephthalate), brominated high impact polystyrene, and polyurethanes have been shown to be biodegradable in a variety of circumstances (as in presence of certain microbial species) despite a strong predisposition suggesting that many of these polymers were recalcitrant to the effects of biodegradation [52].

The average molecular weight of the PLA-based materials before and after fungal degradation was determined by GPC technique, using a refractive index detector. The representative chromatograms are given in Figure 7 and obtained data are summarized in Table 3.

For all samples, it appears a clear tendency to decrease of molecular weight after fungal degradation both for PLA and PLA/R. For PEG-plasticized samples the main peak on chromatograms is split in two with increasing proportion of fractions with low molecular weight. This can be due to loss of PEG which has a low molecular weight and also to degradation. Similar trend was found also in variation of the intrinsic viscosity.

### 3.5. Structural Changes—Fourier-Transform Infrared Spectroscopy (ATR-FTIR) Results

It is well accepted [53] that the biodegradation of PLA is preceded by the chemical hydrolysis of ester bonds in the polymer, which is connected with the decrease of average molecular weight to the level accessible by the enzymatic systems of microorganisms action [38]. The FTIR spectra of PLA, PLA/R, PLA/PEG and PLA/PEG/R before and after being inoculated with *Chaetomium globosum* and *Phanerochaete chrysosporium* fungi are shown in Figure 8a,b.

The general band assignments for PLA-based samples was done according to literature data [25,54,55] and are presented in Table 4. As revealed in our previous studies [29,30,56], neat PLA exhibits sharp bands assigned to vibrations of carbonyl group, ν_C=O_, with a maximum at 1749 cm^−1^ (stretching) and at 1266 cm^−1^ (bending). The absorption bands of native PLA are also noticed in the spectrum of the fungal-biodegraded PLA but slightly shifted to higher wavenumbers and with changes in intensities, and new bands also appears. The spectra of neat PLA, PLA/R, PLA/PEG and PLA/PEG/R samples after being exposed to fungi had a shift in the bands corresponding to C=O group to higher wavenumbers from 1749 cm^−1^ to 1754 cm^−1^ depicting a reduction in their molecular weights. In the case of biodegraded neat PLA sample new bands appear, at 3323, 1652, 1546, 920 cm^−1^, for *Chaetomium globosum* fungus and at 3498, 1654, 1631, 1523, 921 cm^−1^ for PLA exposed to *Phanerochaete chrysosporium*. The mentioned bands above are assigned to -OH stretching, amide C=O stretching, N-H bending and helical backbone vibrations with the CH_3_ rocking modes, respectively. In the case of all biodegraded PLA-based samples, with both fungi, arises in the FTIR spectra a new band at 920–921 cm^−1^ that is characteristic to crystalline phase (namely α-crystals) [57], being accompanied also by a decrease in the intensity of the amorphous band at 955 cm^−1^. This finding indicates that exposure of PLA-based samples to fungal environment determines an increase in crystallinity, which may be assigned to changes in the amorphous/crystalline (more stable to degradation) fraction ratio, being known that the degradation firstly occurs in amorphous region, or to the reorganization of the remaining undegraded chains with lower molecular length because of the molecular weight decrease, which can crystallize [25,58].

For PLA/PEG sample the bands assigned to C–O stretching mode were merged in the very broad envelope centered on 1182 and 1101 cm^−1^ arising from C–O, C–O–C stretches and C–O–H bends vibrations of PEG in PLA. After fungal degradation, first band mentioned is shifted at higher wavenumbers, while the second one is displaced at lower wavenumbers, meaning that oxygen containing bonds suffers changes by fungal action. Furthermore, for the initial samples containing PEG (PLA/PEG and PLA/PEG/R) a characteristic band is observed at 842–843 cm^−1^ that is assigned to C–O stretching, and CH_2_ rocking. After incubation with fungi this band disappears indicating that the addition of the hydrophilic plasticizer leads to an increased susceptibility to biodegradation of the PLA, acting as zones of adhesion for fungi.

FTIR spectra of samples before biodegradation showed the absorbance of CH-COC bonding increased with PEG addition into PLA matrix at frequency 1270 cm^−1^ (a shift to higher wavenumbers, 1279 cm^−1^, being also observed) indicating that binding interaction occurred between PLA and PEG. After incubation with fungi this band suffers a sharp decrease in intensity, revealing that the chemical integrity of the blends is affected. Specific to the samples incubated with the fungi was found also the splitting of the band at around 1453 cm^−1^, attributed to methyl C-H asymmetric bend, which is recently associated with the oxidation and degradation of polymeric materials [59]. 

### 3.6. Thermal Behavior

The glass transition temperature (*T*_g_), melt temperature (*T*_m_), crystallization temperature (*T*_cc_), enthalpy of endothermal melting (Δ*H*_m_) and degree of crystallization (*X*_c_) of samples obtained by DSC technique, before and after fungal degradation were determined.

The DSC curves of PLA and PLA/PEG based samples before and after biodegradation (PLA/14d/CG and PLA/14d/PC) in the presence of the two fungi, PC and CG, were shown in Figure 9 and the values of determined thermal events were summarized in Table 5.

As depicted in Figure 9a, PLA showed the glass transition at 66 °C, cold crystallization temperature at 123.0 °C, which is also present only in DSC curve of the PLA/R sample, and single large melting peak at 152 °C. When the PLA was exposed to the action of the two fungi, very important changes in thermal characteristics were clearly visible. The *T*_g_ values slightly decrease, *T*_cc_ was not more present in DSC curves, while melting peak is split into two melting peaks at 148 °C and 157 °C for PLA/14d/CG and at 150 and 157 °C for PLA/14d/PC, respectively. The two melting peaks observed may result from the melting of two different crystals of PLA which was broken after the action of fungi as the GPC results also proved by decrease of average molecular weight. The shorter chains resulted after degradation can easily reorganize in different crystallites type and this is evident by the two melting peaks in DSC curves and significant increase in crystallinity index *X*_c_ from 11.05% to 32.76% and to 35.70% for PLA/14d/CG and PLA/14d/PC, respectively. A similar melting behavior of PLA was reported by Su et al. [60] and also in our previous paper related to degradation of PLA systems by soil burial [30]. Biodegradation of PLA in the presence of *Chaetomium globosum* was also studied by Ding et al. [61]. The fungi effect on PLA was reported as taking place firstly on the surface and then gradually degraded under fungi environment leading to the decline of tensile and bending properties.

By plasticizing of PLA with PEG, the thermal characteristics resulted in a decrease of *T*_g_ = 44.4 °C and a complex melting peak with two shoulders at 128.9 °C and 144.7 °C and average *T*_m_ = 138.9 °C—Figure 9b. A high crystallinity degree was found of 28.15%, as also reported before in a previous study [56]. A shift in the glass transition was observed after fungi action at 50.3 °C in the case of PLA/PEG/14d/CG and at 59 °C in the case of PLA/PEG/14d/PC, respectively. The single melting peaks remain almost at the same temperature of about 147 °C but the shoulders are not more shown by DSC curves. This could be explain by a simpler composition of the samples after fungal degradation by the loss of PEG and/or the progress of degradation the which also allows the occurrence of the crystallization process and therefore the *X*_c_ significantly increases up to 48%–58%.

The thermal behavior of the PLA/R and PLA/PEG/R samples is similar with those of PLA and PLA/PEG ones, respectively. Finally, the fungal degradation led to increase of the melting in two steps and to the increase in *X*_c_. The biodegraded matrices, PLA/R/14d/CG and PLA/R/14d/PC presented a melting peak split in two distinct melting peaks which may be ascribed to recrystallization of the fragments resulted under fungi action. A similar behavior was found also by Musuc et al. in the case of addition of rosemary to polyethylene films and irradiated [62].

The effect of the fungi on the structure was evidenced by the disappearance of the shoulder and the determined thermal events are shifted indicating presence of smaller molecules which crystalize leading to higher crystallinity. Backes et al. [63] observed a similar phenomenon during the study of PLA/Biosilicate composites and concluded that the crystallization of PLA occurs easier in degraded systems with greater mobility. This outcome can support the hypothesis that after biodegradation in the presence of the two fungi, the chains of PLA/PEG/R became shorter in accordance with GPC results.

### 3.7. Thermogravimetry (TG/DTG) Results

The thermal behavior at high temperatures of PLA matrix before and after biodegradation in the presence of CG and PC fungi was represented in Figure 10 and data are summarized in Table 6.

PLA sample decomposed in a single step between 279–383 °C, with a mass loss of 98%, results being confirmed by previously reported study [57]. PLA samples exposed to the fungi environment showed a lower thermal stability with a mass loss of 91.5% and 95% for CG and PC, respectively.

The plasticized PLA with PEG showed even lower thermal stability by comparison with PLA—Figure 10, PLA/PEG showing a main degradation step between 220–392 °C, where 95% of its mass is lost. PLA/PEG/CG and PLA/PEG/PC showed different degradation profiles. CG seemed to have a stronger effect on the polymeric chains, the product after fungal degradation being more susceptible to thermal degradation.

Addition of rosemary extract to PLA matrices slightly increased the thermal stability, T_onset_ = 282 °C, the DTG peak temperature was found to the same value of 362 °C; T_f_ of PLA/R was found also slightly higher of 390 °C and the degraded mass being of 94%. A small second degradation step was detected at higher temperature over 390 °C which could be assigned to the degradation of the products resulted from first step of degradation. The two fungi used for biodegradation of PLA/R matrices seemed to act in a similar manner on this sample, as no significant differences were observed.

The TG/DTG curves of the PEG-plasticized PLA with incorporated rosemary extract show two degradation steps—Figure 11, the main one occurring between 244–436 °C with a main degradation peak split in one peak and a shoulder at 405 °C; PLA/PEG/R lost during the main degradation step 85% from its mass. A small mass loss of 13.4% occurs at low temperature from 106 °C to 180 °C probably because of desorption of water or of evolved low molecular weight compounds. After fungal degradation, the resulted products showed a much lower thermal stability and a complex mechanism of thermal degradation evidenced by shoulders in DTG curves both before and after main degradation step which take place 160–320 °C with a mass loss of 88%.

## 4. Conclusions

Two fungal strains, *Chaetomium globosum* and *Phanerochaete chrysosporium*, were tested for their ability to biodegrade complex polymeric systems based on PLA and natural bioactive compounds (chitosan and powdered rosemary alcoholic extract). The fungi extracellular enzyme activities (e.g., catalase, superoxide dismutase) are slightly decreased in the case of composites containing natural additives. The presence of natural compounds in the PLA-based polymeric systems determines an acceleration of fungal degradation and probably the chemical hydrolysis, which further helps the attachment of fungi to the resulting polymeric fragments and consequently the acceleration of biodegradation. Significant decreases in average molecular mass of the polymeric samples were observed by fungi action; accompanied by structural changes, increase in crystallinity and decrease of thermal properties, the loss of the physical integrity and finally to degradation with integration in environmental medium. Such findings are in accordance with other reports on microbial degradation of polymeric materials which offer new emerging technological opportunities to treat the enormous pollution problems arising by the use of polymeric products. Degrading fungi (some found in soil), as *Aspergillus tubingensis* [64], showed the ability to biodegrade various polymers. Both fungi tested here proved to be efficient for PLA-based material degradation, the best from them being *Chaetomium globosum.*

## Figures and Tables

**Figure 1 polymers-12-00469-f001:**
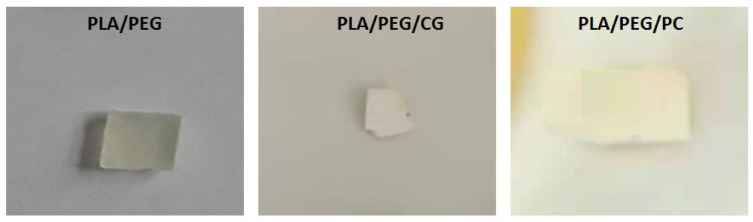
Visual appearance of PLA/polyethylene glycol (PEG) sample before and after 7 days exposure to *Chaetomium globosum (CG)* and *Phanerochaete chrysosporium (PC)* fungi.

**Figure 2 polymers-12-00469-f002:**
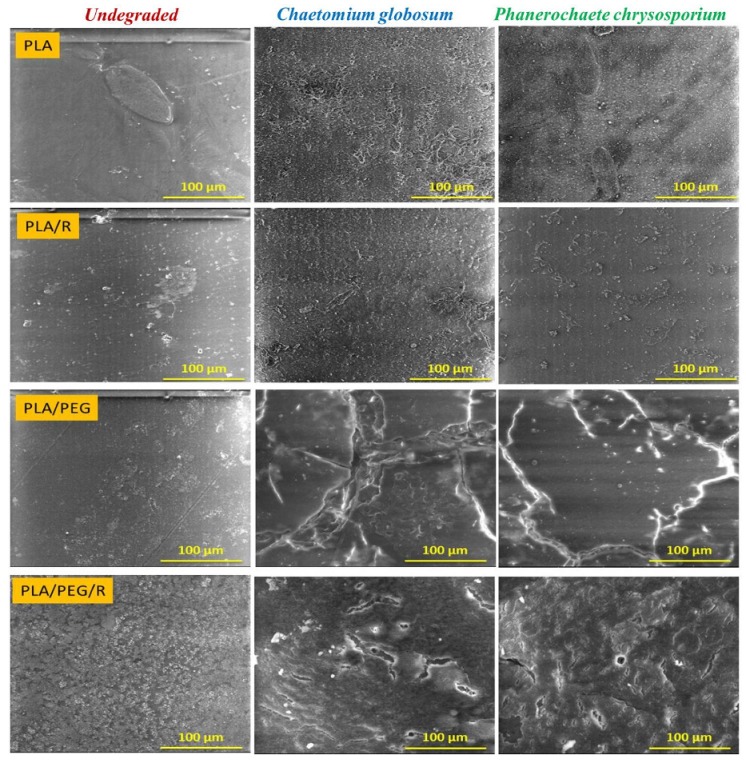
SEM micrographs (magnification of 1000×, scale bar—100 µm) of PLA-based samples after 14 days in *Chaetomium globosum* and *Phanerochaete chrysosporium* fungi comparatively with undegraded ones.

**Figure 3 polymers-12-00469-f003:**
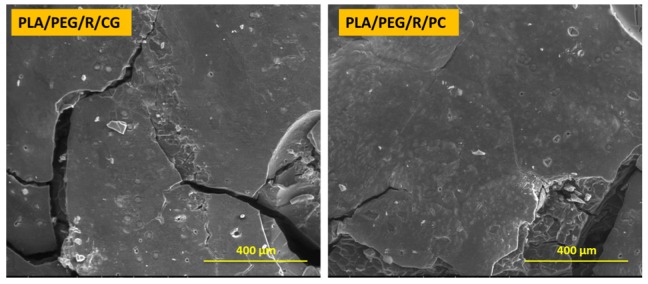
SEM micrographs (magnification of 250×, scale bar—400 µm) of PLA/PEG/R sample after 14 days incubation with *Chaetomium globosum* (CG) and *Phanerochaete chrysosporium* (PC) fungi.

**Figure 4 polymers-12-00469-f004:**
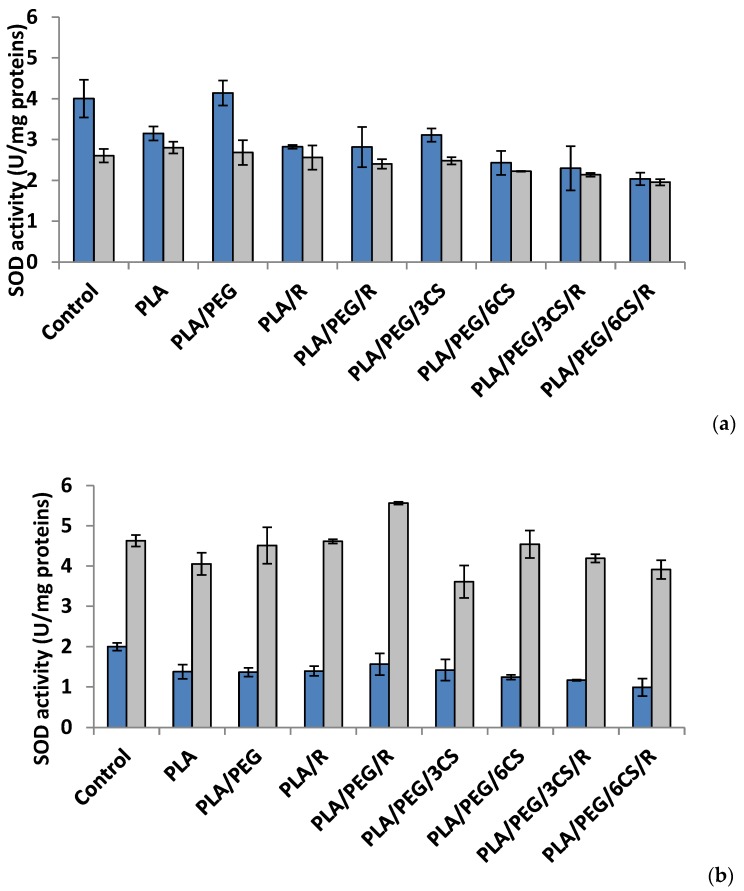
Superoxid dismutase (SOD) activity in *P. chrysosporium* (dark blue columns) and *C. globosum* (grey columns) after 7 days (**a**) and 14 days (**b**) of incubation in the presence of PLA-based samples.

**Figure 5 polymers-12-00469-f005:**
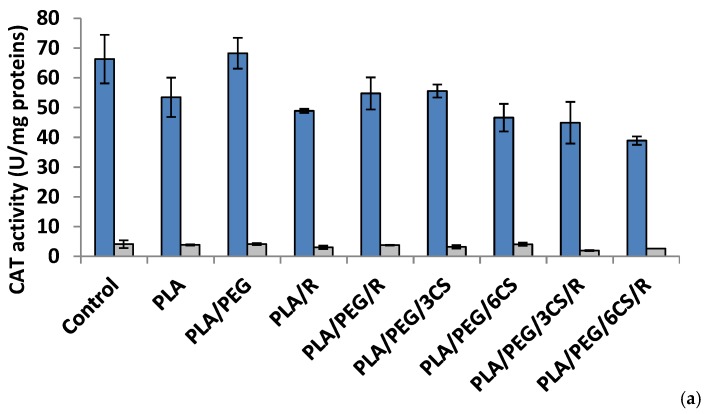
Catalase (CAT) activity in *P. chrysosporium* (dark blue columns) and *C. globosum* (grey columns) after 7 days (**a**) and 14 days (**b**) of incubation in the presence of PLA-based samples.

**Figure 6 polymers-12-00469-f006:**
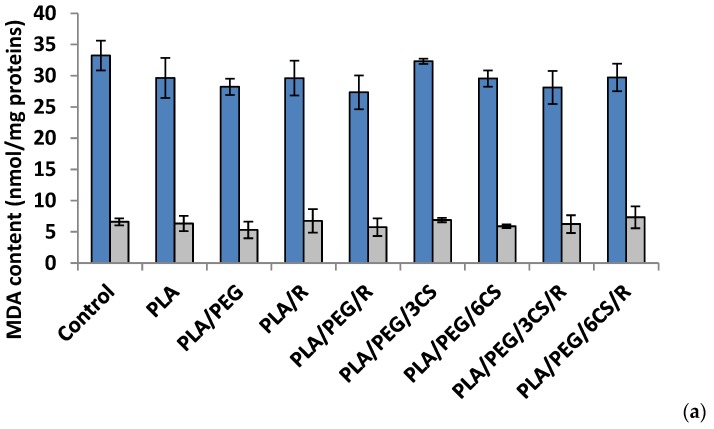
Malondialdehyde (MDA) content in *P. chrysosporium* (dark blue columns) *and C. globosum* (grey columns) after 7 days (**a**) and 14 days (**b**) of incubation in the presence of PLA-based samples.

**Figure 7 polymers-12-00469-f007:**
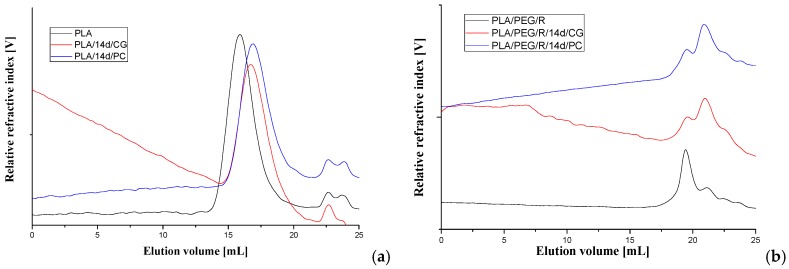
GPC chromatograms of (**a**) PLA and (**b**) PLA/PEG/R samples before and after 14 days exposure to *Chaetomium globosum* (CG) and *Phanerochaete chrysosporium* (PC) fungi.

**Figure 8 polymers-12-00469-f008:**
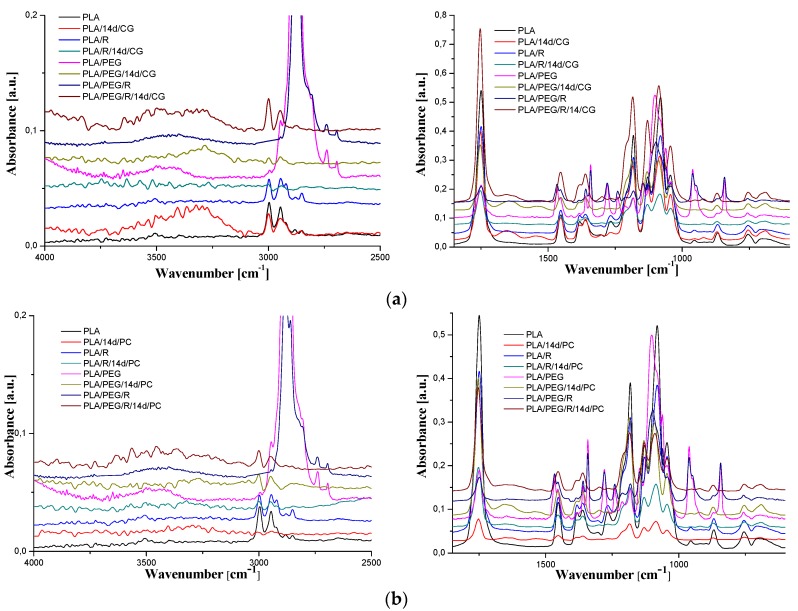
FTIR spectra of PLA-based samples before and after 14 days fungal exposure to: (**a**) *Chaetomium globosum* and (**b**) *Phanerochaete chrysosporium*.

**Figure 9 polymers-12-00469-f009:**
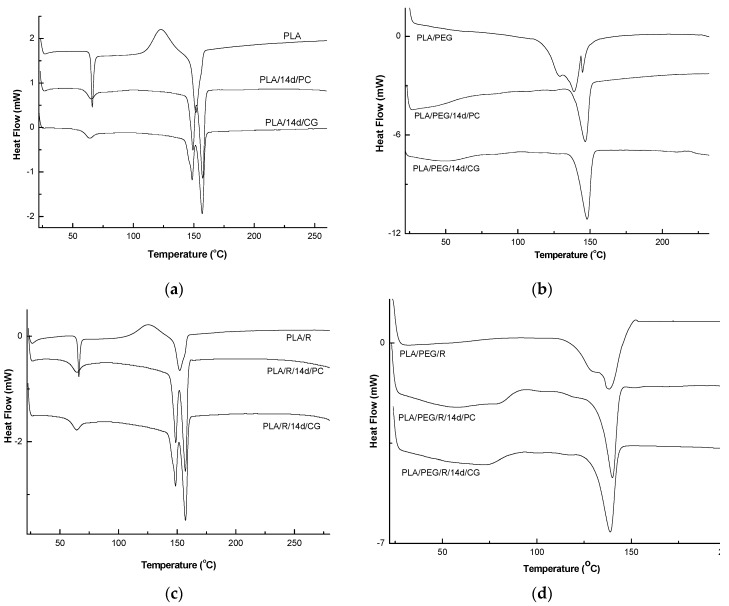
Differential scanning calorimetry (DSC) curves of undegraded and biodegraded PLA (**a**), PEG-plasticized PLA (**b**); PLA/R (**c**); PLA/PEG/R (**d**).

**Figure 10 polymers-12-00469-f010:**
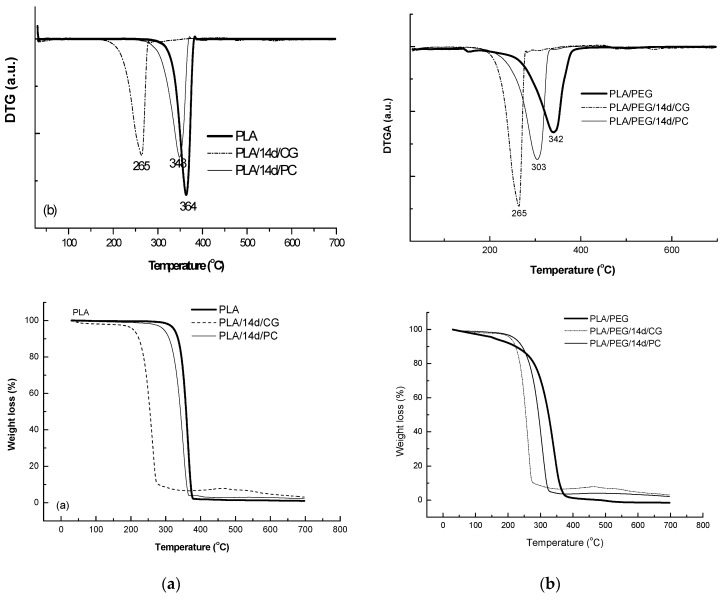
Thermogravimetry (TG/DTG) curves of PLA (**a**) and PLA/PEG (**b**) based systems before and after biodegradation.

**Figure 11 polymers-12-00469-f011:**
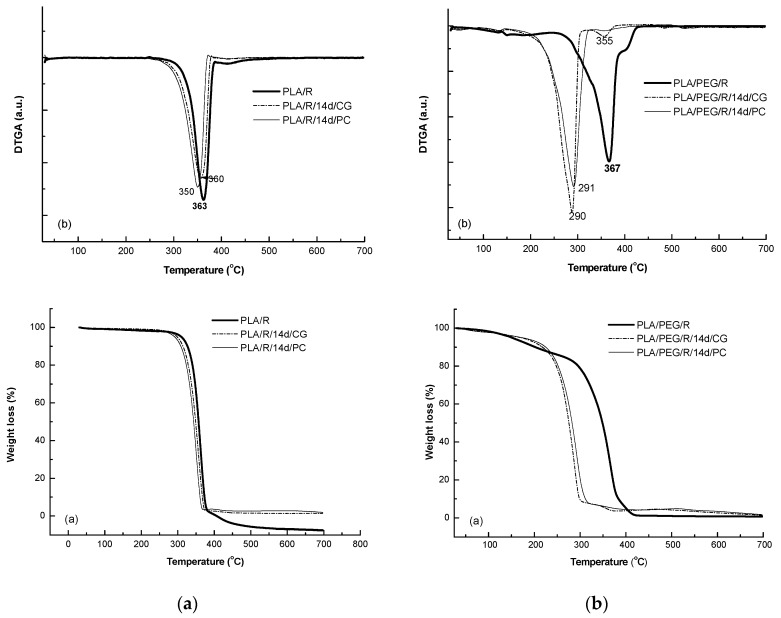
TG/DTG curves of PLA/R (**a**) and PLA/PEG/R samples before and after biodegradation (**a**) TG curves and (**b**) first derivative DTG curves.

**Table 1 polymers-12-00469-t001:** Poly(lactic acid) (PLA)-based samples that were exposed to fungal biodegradation.

Samples Description and Codes		
Before Biodegradation	After Biodegradation (7 or 14 days)	
	*Chaetomium globosum (CG)*	*Phanerochaete chrysosporium (PC)*
Native poly(lactic acid) (PLA)	PLA/7d/CG or PLA/14d/CG	PLA/7d/PC or PLA/14d/PC
PLA plasticized with polyethylene glycol (PLA/PEG)	PLA/PEG/7d/CG or PLA/PEG/14d/CG	PLA/PEG/7d/PC or PLA/PEG/14d/PC
PLA containing powdered rosemary ethanolic extract (R) 0.5% (PLA/R)	PLA/R/7d/CG or PLA/R/14d/CG	PLA/R/7d/PC or PLA/R/14d/PC
PLA plasticized with PEG containing powdered rosemary ethanolic extract (PLA/PEG/R)	PLA/PEG/R/7d/CG or PLA/PEG/R/14d/CG	PLA/PEG/R/7d/PC or PLA/PEG/R/14d/PC
PLA/PEG containing 3wt% chitosan (PLA/PEG/3CS)	PLA/PEG/3CS/7d/CG or PLA/PEG/3CS/14d/CG	PLA/PEG/3CS/7d/PC or PLA/PEG/3CS/14d/PC
PLA/PEG containing 6wt% chitosan (PLA/PEG/6CS)	PLA/PEG/6CS/7d/CG or PLA/PEG/6CS/14d/CG	PLA/PEG/6CS/7d/PC or PLA/PEG/6CS/14d/PC
PLA/PEG/R containing 3wt% chitosan and powdered rosemary ethanolic extract (PLA/PEG/3CS/R)	PLA/PEG/3CS/R/7d/CG or PLA/PEG/3CS/R/14d/CG	PLA/PEG/3CS/R/7d/PC or PLA/PEG/3CS/R/14d/PC
PLA/PEG/R containing 6wt% chitosan powdered rosemary ethanolic extract (PLA/PEG/6CS/R)	PLA/PEG/6/CS/R/7d/CG or PLA/PEG/6CS/R/14d/CG	PLA/PEG/6/CS/R/7d/PC or PLA/PEG/6CS/R/14d/PC

**Table 2 polymers-12-00469-t002:** The percent of weight loss of PLA-based samples subjected to fungal cultures.

Sample	*Weight Loss (%)*			
	*Chaetomium globosum*		*Phanerochaete chrysosporium*	
	7 days	14 days	7 days	14 days
PLA	18	22	9	18
PLA/PEG	43	57	42	49
PLA/R	31	42	25	33
PLA/PEG/R	42	49	40	47
PLA/PEG/3CS	100	100	100	100
PLA/PEG/6CS	100	100	100	100
PLA/PEG/3CS/R	100	100	100	100
PLA/PEG/6CS/R	100	100	100	100

**Table 3 polymers-12-00469-t003:** Average number (Mn), weight (Mw) and Z average (Mz) molecular weight, polydispersity (Mw/Mn, Mz/Mw) and intrinsic viscosity [η] of the PLA-based samples before and after 14 days of fungal degradation.

Sample	Mn × 10^2^ g/mol	Mw × 10^2^ g/mol	Mz × 10^2^ g/mol	Mw/Mn	Mz/Mw	[η] mL/g × 10^2^
PLA	1150	2170	3670	1.878	1.695	1.447
PLA/14d/CG	626	1000	1540	1.598	1.539	0.739
PLA/14d/PC	526.2	889.0	1404	1.690	1.579	0.688
PLA/PEG	92.2	110.2	138.2	1.169	1.254	0.061
	38.46	44.28	49.56	1.151	1.119	0.131
PLA/PEG/14d/CG	87.1	103.2	123.1	1.183	1.193	0.06
	44.18	47.01	49.99	1.064	1.063	0.188
PLA/PEG/14d/PC	84.4	88.48	92.94	1.049	1.050	0.012
	39.87	43.79	31.29	1.098	1.101	0.619
PLA/R	1060	1788	2864	1.687	1.602	1.293
PLA/R/14d/CG	456.2	768.0	1209	1.683	1.574	0.635
PLA/R/14d/PC	374.1	635.3	984.9	1.698	1.550	0.567
PLA/PEG/R	95.20	113.0	140.8	1.187	1.247	0.071
	39.40	44.85	50.03	1.138	1.115	0.016
PLA/PEG/R/14d/CG	88.54	92.92	97.61	1.050	1.050	0.019
	49.64	51.94	54.49	1.046	1.049	0.019
PLA/PEG/R/14d/PC	86.29	91.26	96.39	1.058	1.056	0.382
	44.36	47.05	50.28	1.061	1.069	0.0058

**Table 4 polymers-12-00469-t004:** FTIR spectroscopy data and the band assignment.

Bands Wavenumber (cm^−1^)												Band Assignment
PLA			PLA/PEG			PLA/R			PLA/PEG/R			
Undegraded	Degraded		Undegraded	Degraded		Undegraded	Degraded		Undegraded	Degraded		
	*CG*	*PC*		*CG*	*PC*		*CG*	*PC*		*CG*	*PC*	
-	-	3498	3437	-	-	3511	3518	-	-	-	3564	ν(OH)free
-	3323	-	-	3282	3271	-	-	3350	3399	3402	3359	ν(OH) H-bonded
2996	2999	3001	-	2997	3001	2997	3001	2999	-	2999	3001	ν_as_CH_3_
2946	2947	2947	2945	2947	2947	2946	2943	2947	2945	2947	-	ν_as_CH_3_
-	-	-	2884	-	-	-	-	-	2884	-	-	νCH
1749	1753	1754	1749	1754	1754	1750	1755	1754	1749	1754	1754	ν(C=O)
-	1652	1654	-	1647	1654	-	1666	1635	-	1654	1627	Amide(C=O)
-	-	1631	-	-	-	-	-	1584	-	-	-	νNH
-	1546	1523	-	1542	1541	-	1523	1519	-	1527	1517	νNH
-	-	-	**1467**	-	-	-	-		**1467**	-	-	CH bending
1452	1452	1452	1454	1452	1452	1452	1452	1452	1454	1454	1454	δ_as_CH_3_
-	-	-	1413	-	-	-	-	-	1416	-	-	
1381	1385	1382	1383	1382	1384	1382	1379	1382	1384	1385	1388	δ_s_CH_3_
1361	1359	1361	1359	1359	1359	1360	1359	1361	1359	1361	1361	*δl*CH + δ_s_CH_3_
-	-	-	**1341**	-	-	-	-	-	**1341**	-	-	δ_s_CH_3_
1303	1299	1299	1279	1298	1298	1302	1299	1298	1280	1298	1298	δ2CH
1266	1267	1267	1240	1267	1267	1267	1267	1265	1240	1267	1265	δCH + νCOC
1211	1209	1211	1212	1209	1209	1210	1209	1210	1211	1209	1210	ν_as_COC
1182	1184	1184	1182	1186	1186	1182	1184	1184	1181	1184	1186	ν_as_COC
1129	1130	1130	1146	1130	1130	1129	1130	1130	1146	1130	1130	r_as_CH_3_
1081	1083	1085	1101	1089	1091	1081	1083	1085	1097	1087	1089	ν_s_COC
-	-	-	1061	-	-	-	-	-	1060	-	-	νCOC
1044	1043	1045	1044	1043	1045	1044	1043	1045	1042	1043	1043	νC–CH_3_
954	955	954	961	954	954	954	952	954	961	952	953	rCH_3_ + νCC
-	920	921	948	920	921		921	921	949	921	921	rCH_3_ + νCC
867	869	869	870	869	869	868	869	869	871	869	869	νC–COO
-	-	-	843	-	-	-	-	-	842	-	-	νC-O + rCH_2_
755	754	756	757	754	754	755	754	754	757	754	754	δC=O
692	690	692	694	690	690	700	692	692	692	690	692	γC=O

CG—*Chaetomium globosum*, PC—*Phanerochaete chrysosporium*, s—symmetrical and as—asymmetric; ν—in plane vibration; γ—out of plane vibration; r—rocking vibration; δ_s_—scissoring vibration.

**Table 5 polymers-12-00469-t005:** Thermal characteristics of the PLA based systems before and after fungal degradation determined by DSC.

Samples Code	*T*_g_ (°C)	*T*_cc_ (°C)	*T* _m_	Δ*H*_m_ (J/g)	*X*_c_ (%)
PLA	66	123.0	151.8	−10.23	11.05
PLA/14d/CG	64.1		Two peaks 148.7 156.9	−12.13 −18.37	32.76
PLA/14d/PC	65.3		Two peaks 149.3 157.3	−13.98 −19.26	35.70
PLA/PEG	44.4		128.9sh 138.9 144.7sh	−26.21	28.15
PLA/PEG/14d/CG	50.3		147.9	−53.76	57.73
PLA/PEG/14d/PC	46.5		146.6	−44.39	47.95
PLA/R	66.2	125.4	152.2	7.81 −13.09	14.06
PLA/R/14d/CG	64.4		Two peaks 148.6 157.1	−13.73 −17.53	33.57
PLA/R/14d/PC	64.4		Two peaks 148.7 157.0	−18.74 −17.69	39.13
PLA/PEG/R	48.4		129.0sh 138.3	−12.03	12.92
PLA/PEG/R/14d/CG	49.5		138.8	−61.43	65.98
PLA/PEG/R/14d/PC	46		Two peaks 116.0 139.9	−49.52	53.19

**Table 6 polymers-12-00469-t006:** Thermal characteristics of the PLA and PLA/PEG based systems determined by TG.

Samples Code	TG/DTG Results			
	T_onset_ (°C)	T_peak_ (°C)	T_f_ (°C)	Δm (%)
PLA	279	362	383	98.3
PLA/14d/CG	235	357	384	91.52
PLA/14d/PC	248	350	367	95
PLA/PEG	152	342	392	95
PLA/PEG/14d/CG	156	265	327	90.33
PLA/PEG/14d/PC	157	303	367	93.9
PLA/R	282	362	390	94.0
	393	417s	438	
PLA/R/14d/CG	251	356	382	95.66
PLA/R/14d/PC	243	352	371	94.75
PLA/PEG/R	106	146	180	13.43
	244	364; 405sh	436	84.87
PLA/PEG/R/14d/CG	108	132	149	2.27
	160	287	316	88.6
	325	355	382	4.99
	455	482	507	3.08
PLA/PEG/R/14d/PC	111	143	143	5.35
	160	292	325	87.66
	343	358	388	2.95
	507	527	559	0.89

T_onset_—onset temperatures, T_peak_—temperature corresponding to maximum mass loss rate, T_f_—final temperature of process and Δm—mass loss at the end of each process); sh—shoulder.

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
