# Peer review of "Influence of the Chitosan and Rosemary Extract on Fungal Biodegradation of Some Plasticized PLA-Based Materials"

_polymers, 2020, doi:10.3390/polym12020469_

Round 1
Reviewer 1 Report
In the Introduction the definition and application of plant extract/botanicals should be added and some references in this regards.
Durazzo et al. Polyphenols: A concise overview on the chemistry, occurrence, and human health. Phytother Res. 2019 Sep;33(9):2221-2243.
Durazzo et al.From Plant Compounds to Botanicals and Back: A Current Snapshot. Molecules. 2018 Jul 24;23(8). pii: E1844. doi: 10.3390/molecules23081844.
The aim of paper should be clarified and rewritten.
In paragraph 2.4.1. Scanning Electron Microscopy (SEM), the advantages of SEM should be marked.
Table 2 should better described in the text.
The results in figure 6 should be better described and discussed in the text.
At line 271 the authors should give a definition of antioxidant compound and describe the update research on antioxidant properties and related references should be added.
Table 3 and Table 4 should be redesigned.
Author Response
We are sending you the corrected manuscript of the paper entitled: Influence of the Chitosan and Rosemary Extract on Fungal Biodegradation of Some plasticized PLA-Based Materials by Elena Stoleru, Cornelia Vasile, Lăcramioara Oprică, Onur Yilmaz to be published if accepted in your prestigious journal: Polymers in special issue ""Natural Additives for Special and High Performance Polymeric Materials".
We thanks to the reviewers for their valuable comments which helps to improve the quality of the manuscript.
Thanking for your fine collaboration.
Best Regards
Dr. Cornelia Vasile

Reviewer 2 Report
Please check the affiliation part. The first affiliation contains short form “Blvd” and the second affiliation does not contain the full name of the university (lines 9 and 11).
Line 17 - There is no space between words (chrysosporiumwere).
Please modify the sentence “Humidity, 60°C temperature, and microorganisms are required for PLA biodegradation, conditions which can be found sometime in garden soil.”
From line 55 to line 69 it contains different color highlights.
Line 78 – “14 filamentous fungal…” I recommend changing the sentence and not starting with a number.
Line 80 “PLLA”. What is PLLA? It is not defined.
Line 86 “It was showed [21] that…” What is the name of the person who demonstrated that use?
Line 90 “Chaetomium globosum” Different font.
Line 104 “wt%” - The abbreviation name is missing. In addition, the introductory part is missing the most important topic, chitosan; its properties have not been discussed.
Line 133 “…on Erlenmeyer flask with 100 ml Haynes liquid or ml Sabouraud liquid media…” I recommend that you complete this sentence.
Author Response

(The authors gave the same response as above.)
